# Why women do not use contraceptives: Exploring the role of male out-migration

**Saradiya Mukherjee, Bidhubhusan Mahapatra** **\*, Niranjan Saggurti**

Population Council, New Delhi, India

\* bbmahapatra@popcouncil.org, bbmahapatra@gmail.com

## Abstract

### Introduction

Contextual understanding of reasons for non-use of contraceptives is key to devising family planning (FP) strategies. This study aimed to understand the reasons for non-use of contraceptives among women in the context of male out-migration.

### Methods

Focus Group Discussions (FGDs) were conducted in two high male out-migration districts in Bihar, namely Nawada and Gopalganj. Twenty-five FGDs were conducted with currently married women with migrant husbands, currently married men and Accredited Social Health Activists (ASHAs) during April-June 2019. Data were analysed by using a thematic approach through Atlas.ti-6.2.

### Results

The reasons for contraceptive non-use in areas with high male out-migration were complex, as this included barriers to contraceptive use unique to high-migration areas and reasons commonly found in other rural settings. Non-use of contraceptives among women with migrant husbands was mostly driven by lack of contraceptive preparedness before husband's arrival, inability to procure contraceptives due to inaccessibility to health facilities and stigma to procure when husband was away. Other migration environment related factors included low ASHA outreach, myths and side effects of contraceptives, community fertility norms and poor spousal communications around FP.

### Conclusion

The reasons for non-use of contraceptives are multifaceted, complex and interlinked. Exploration of these reasons in migration context suggest that FP programs in migration affected areas need to address a range of barriers to contraceptive use at multiple levels.

**Data Availability Statement:** The data underlying the results presented in the study cannot be made publicly available because they contain sensitive, identifying information. Data access queries may,

however, be sent to Mr. Ashutosh Mishra, (contact via ashutosh@popcouncil.org), the institutional contact for all data collected by Population Council India office.

**Funding:** Data for this study was collected with support from the David and Lucile Packard Foundation. The views expressed herein are those of the authors and do not necessarily reflect the official policy or position of the David and Lucile Packard foundation. The funders had no role in the study design, data collection and analysis, decision to publish, or preparation of the manuscript.

**Competing interests:** The authors have declared that no competing interests exist.

## Introduction

India has made considerable efforts over the last few decades to create a conducive family planning (FP) environment where couples can make conscious decisions on timing of births, spacing between births or limiting births through contraceptive use to achieve desired family size by taking several important policy and programmatic decisions [1–3]. During this journey multiple strategic approaches have been adopted, including a target specific approach, contraceptive-specific incentives, and FP camp approach [2,3]. Nonetheless, with over half of its population in the reproductive age group and the majority of its population living in villages, opportunities for FP progress in India are abundant but come with lots of challenges [1]. Despite decades of investments made in FP programs in India, a significant proportion of women, particularly, in states like Bihar and Uttar Pradesh, do not use modern contraceptive methods [4]. Moreover, a decline in contraceptive use was noted in the Bihar in the last decade. Even among those who used modern contraceptives, female sterilization was the most dominant method. Of the 86% women who were sterilized, 44% had it before they turned 35 years old [4]. Empirical research in Bihar and other parts of India suggests a variety of reasons for non-use of contraceptives, such as side effects, difficulty in accessing FP methods, and husband's opposition [4–7]. While these reasons are more or less true across various population groups in the country, they need to be examined with proper contextual understanding to have more nuanced understanding of the FP behaviours and decision making among women and couples.

Migration can be one such contextual factor that may help understand the reasons for non-use in Bihar. Empirical research examining effects of migration on various health issues has indicated that male out-migration is negatively associated with utilization of reproductive health services and contraceptive use among women in developing countries [8–12]. A recent study in Nepal has shown that women with migrant husbands had significantly lower contraceptive use than women who had non-migrant husbands [13]. Some of these studies have also shown that migration of husbands leads to poor spousal communication on FP issues, which adversely affects use of contraceptives [10,11,14]. In Bihar, specifically, there has been no research examining FP issues in the context of migration, though there have been studies examining effects of male out-migration on HIV, and utilization of maternal health services. Studies also reported significantly higher prevalence of HIV among men with a history of migration and among women, whose husband had a history of migration [15–18]. One of these researches has shown that migrating men carried HIV infection from destination areas (high HIV prevalence states) to rural areas of Bihar [19]. A study conducted in Bihar to understand the impact of male out-migration on the reproductive health status of the wives left behind, showed that wives left behind by migrants were more likely to have reported the symptoms of reproductive morbidity compared to the wives of non-migrants [20]. Given this history of linkages between migration and health behaviours in Bihar, it is important to examine how FP behaviours are influenced by migration. Specifically, how do reasons for contraceptives non-use could be viewed from the lens of migration. In this backdrop, this study aimed to understand the reasons related to contraceptives non-use in the context of male out-migration in Bihar and explored how the reasons interacted with each other.

## Methods

### Study setting

The study was conducted in two high male out-migration districts of Bihar: Gopalganj and Nawada. The two districts were chosen as they have very high levels of male out-migration

(Nawada: 36%, Gopalganj: 33% [4]), as well as one represents northern Bihar (Gopalganj) and the other is located in southern Bihar (Nawada). According to the 2011 census, major out-migration destinations for men from Nawada were West Bengal, Jharkhand, and Uttar Pradesh; whereas, Maharashtra, Gujarat and Punjab remained major migration destinations for men migrating from Gopalganj. Long distance migrant returns once in a year whereas short distance ones return more frequently. Most of the migrant men return to their native place during key festival seasons such as *Chhath, Holi and Dushera.* Depending on the type of migrant, long or short distance, husbands stay for 15–45 days when they visit home. Socio-economically, these two districts were similar. Almost half (47%) of the women in Nawada and slightly lower than one-third (57%) women in Gopalganj were found to be literate [4]. More than two-fifths of the household in both the districts (44% in Gopalganj, 47% in Nawada) belonged to the poorest wealth quintile [4]. The use of modern contraceptives was lower in Gopalganj (9%) than in Nawada (29%) [4].

## Study design

Qualitative data collection in the form of focus group discussions (FGD) was conducted during April-June 2019. FGDs were conducted to understand the perspectives on contraceptive use and FP in communities affected by migration. Compared to in-depth interviews, FGDs were preferred to bring out divergent views which helped to understand the community all together. The other objective of conducting FGDs was to inform the quantitative survey in developing study tools tuned for the context of study. In each district, four blocks (two high male out-migration and two low male out-migration) were selected. The blocks were selected after consulting with the district statistical office and district labour office, that keep track of male out-migration from the districts. Based on the information provided, blocks in the districts were arranged in ascending order of volume of male out-migration. The top two and bottom two male out-migration blocks were selected from the list. Next, the study team visited the selected blocks and conducted key informant interviews with 4–5 individuals at block level to identify village clusters (groups of 4–5 villages) with high volume of male out-migration. Two such clusters from each block were selected for conducting FGDs. A total of 25 FGDs were conducted with currently married women with migrant husbands, currently married men (both migrant and resident) and Accredited Social Health Activists (ASHA) across the selected blocks of the study districts. The study conducted one FGD each with women and ASHAs in each study block. Similarly, for men, one FGD was conducted in each block except for one where separate FGDs were conducted for migrant and non-migrant men.

**Selection of women and men.** The study team planned to conduct at least one FGD with each respondent group in each block. Overall, eight FGDs were conducted covering 70 currently married women. Similarly, nine FGDs were conducted with 77 currently married men. On an average, woman were 27 years old and men were 32 years old (Table 1). For selection of respondents, lists of eligible women (currently married, aged 15–45 years, have migrant husband) and eligible men (currently married and aged 15–45 years) were prepared in

**Table 1. Sample size and profile of FGD participants.**

| Type of participants | Women | Men | ASHA | Total |
|---|---|---|---|---|
| Number of FGDs | 8 | 9 | 8 | 25 |
| Number of FGD participants | 70 | 77 | 72 | 219 |
| Mean age (years) | 26.5 | 31.8 | 38.5 | – |
| Mean years of schooling completed | 4.3 | 8.1 | 8.6 | – |

consultation with village heads and ASHAs in each village. The study team then approached the eligible participants and requested them to participate in the study. Individuals who provided their verbal consent were asked to reach the venue where the FGD was scheduled to take place.

**Selection of ASHA.** One FGD was conducted in each of the selected block. Each selected block had one Primary Health Center (PHC). Around 80 ASHAs worked under one PHC. In order to select ASHAs for FGDs, ASHA facilitators were contacted for each PHC and the list of ASHAs currently working was obtained. From the list, 10 ASHAs were selected randomly and contacted for their participation in the FGDs. Those who verbally agreed to participate were asked about their availability on the upcoming ASHA divas (monthly meeting of ASHAs working under a PHC conducted at the PHC). As ASHAs belonged to different villages, FGDs were conducted on those days when PHCs conducted ASHA divas. A total of eight FGDs were conducted consisting of 72 ASHAs (Table 1).

**Interview procedure.** All the respondents who provided verbal consent, were again explained about the study procedures, objectives, harms, benefits and voluntariness of their participation at the interview venue. After that, a written consent was taken from all respondents. A copy of the signed consent form was given to the respondents as well. All the FGDs were conducted in private places. The study rented a place in the village to conduct the interviews with men and women. The ASHA interviews were conducted in a room in the PHC compound. Women FGDs were conducted in women-only environment, men FGDs were conducted in a men-only environment and all the interviews were conducted by gender-matched research investigators. For FGDs with men and women, semi-structured guides focussing on participants' perception of contraceptive methods, social norms, use of contraception, communication and decision-making on contraceptive use were used. No monetary compensation was provided to participants of the study. The discussion guides for AHSAs were focused on perception in community about contraceptive methods, social norms on fertility and contraceptive use, ASHAs' role and challenges in FP service delivery, decision making and use of contraceptive methods among migrant and non-migrant households. The FGDs were facilitated by individuals whose native language was Hindi and had a graduation degree in a social science subject with prior experience of conducting qualitative and quantitative interviews. For conducting FGDs, study teams were formed where each team had three members: one each for facilitating the discussion, taking notes and coordinating with respondents, and ensuring privacy. In addition to the note taking, interviews were also audio-recorded. At the end of each interview, the study team listened to the audio and prepared the final notes in Hindi. The notes were later translated into English for coding and analysis.

## Ethical approval

Approvals for the study were obtained from the Institutional Review Board of the Population Council. Written consent was taken from all study participants.

## Data analysis

All textual data were analysed by using an inductive and iterative process. Initial codes were developed according to predetermined categories and organized into various themes and sub-themes using a thematic approach [21]. Next, the coding lists were expanded and further reviewed and summarised. A summary of categories was developed for every category of respondents by contrasting and synthesizing information across all themes and subthemes. All data from the translated notes from women, men and ASHAs were coded and analysed separately. Coding was done by two researchers independently. After the coding were completed,

they were compared, and a consolidated coded file was created. Analysis was performed in Atlas.ti-6.2, a computer-based text search program that allows multiple codes to be searched and linked simultaneously.

## Results

### Contraceptive use

Contraceptive use, specifically of modern methods was low in the study community. Both men and women regardless of their age and parity, were reluctant to use contraceptives. Majority of participants across all interviews alluded that FP was not a matter of concern for them; more-over, they were very eloquent in opposing family planning.

> *"FP is very bad. I had two sons and a daughter, after birth of my daughter my wife underwent an operation [tubectomy]. Last year my daughter passed away, tell me if I want to have a daughter now, how I can undo the operation [tubectomy]*–Married resident man, Age 39, FGD1, High male-out migration area.

> *"I have never used any FP method. We do not keep sexual contact at the time of menstruation for 10–15 days if my husband visits home during that time. For the rest, [we have sex] without using anything. We keep control on ourselves."*- Woman with migrant husband, Age 20, FGD 2, High male-out migration area.

> *"Using FP methods is not good for health, that is why we do not prefer to use. Self-control is the best form of contraceptive method."*–Married migrant man, Age, 31, FGD 3, High male-out migration area.

In this community, children are considered as god's gift which manifests into the popular notion in that area that one should have as many children as they want, number does not mat-ter. The idea of FP and contraceptive use is not considered important, as it appeared during the discussions.

> *"Use of method [contraceptives] is very low in villages. Many women [in this area] still say that children are given by God. If God gives a child, why to stop that by using anything [con-traceptives]."*–Woman with migrant husband, Age 25, FGD 2, High male-out migration area.

Analysis of the reasons for contraceptive non-use suggested that they can be grouped into two broad areas: (i) reasons pertaining to husband's out-migration, and (ii) reasons pertaining to migration environment. The factors identified (Table 2) had multifaceted influence on

**Table 2. Reasons for non-use of contraceptives among currently married women and men in Bihar.**

|  | Reasons for non-use of contraceptives |
|---|---|
| **Reasons pertaining to husband's migration** | Low perceived need for contraceptive methods<br>Lack of preparedness to procure contraceptives during husband's visit<br>Fear of side effects<br>Inability to procure contraceptive methods<br>Pregnancy as contraceptive |
| **Reasons pertaining to migration environment** | Low ASHA outreach Preference of male child<br>Need to prove fertility<br>Myths and misconceptions about contraception<br>Poor couple communication on FP |

women and men's decision to use or not use contraceptives. In the following section, each of these reasons for non-use are detailed out.

### Reasons pertaining to husband's migration

A majority of respondent agreed that women did not use contraceptives when husbands were away. Non-use of contraceptives among women with migrant husbands was mostly driven by their own perception and the experience of other women in the community.

**Low perceived need for modern contraceptive methods.** Given that husbands stayed in a different location, most of the women did not feel the need for using contraceptives. Husband's migration was a form of natural contraception for them. Nevertheless, it was not only women who perceived the low need, even ASHAs discouraged them from using any contraception. Women during the discussion narrated that ASHAs advised them to procure contraceptives at the time of husbands' return. Households where the husband was usually away for work for a long period of time were not considered mainstream users of contraceptives by the ASHAs. ASHAs only considered it relevant to talk to a woman with a migrant husband about contraceptive use only when her husband was at home.

> *"Why to use any method when husbands are away, when they come, we refrain from having sex on the specific days [on unsafe days]."*- Woman with migrant husband, Age 23, FGD 7, High male-out migration area.

> *"Some women with migrant husband ask about what contraceptive method they should use, but I tell them, not to worry and ask them to meet us when husbands come back."*- ASHA, Age 35, FGD 5, High male-out migration area.

**Lack of preparedness to procure contraceptives during husband's visit.** The low perceived need for contraceptive use might have led to poor preparedness to procure contraceptive methods before husband's arrival. Women who intended using contraceptives waited until husbands' arrival. However, once husbands arrived home women got busy and did not get time to procure contraceptives from PHCs or ASHAs. In some circumstances when migrant husband came home unannounced women could not use any contraceptive method.

> *"Some women with migrant husbands would wait until their husbands reach home. After that, these women do not get time to consult ASHAs or get opportunity to procure methods."*—Woman with migrant husband, Age 30, FGD 7, High male-out migration area.

> *"When husband is away women do not use anything and if husband comes back without informing, then they have no methods with them to use."* ASHA, Age 38, FGD 8, High male-out migration area.

**Fear of side effects.** Women with migrant husbands were concerned that using contraceptives may result in side effects and they may fall sick. Commonly reported side effects included headache, body ache, disturbance in menstruations, weight gain/loss, nausea caused by oral pills, and other medical problems after long use of intrauterine devices (IUD). In some cases, women complained about stomach pain, unusual heavy bleeding after insertion of IUD or after usage of injectables (called *Antara* injection in Bihar). The fact that women were the primary caregivers for their children and were responsible for household duties during husband's absence, they were worried about who would take care of their children and other household responsibilities if they fell sick by using contraceptive methods.

*"Some women are afraid to try contraceptives. They are afraid to fall sick. If they fall sick who will take care of all the children when their husbands do not stay at home?"*—Woman with migrant husband, Age 27, FGD 6, High male-out migration area.

**Inability to procure contraceptive methods.** For multiple reasons, accessing contraceptive methods was difficult for a woman when her husband was away. Women expressed their inability to travel to PHCs alone. Women in the study areas were allowed to visit limited places and were mostly accompanied by escorts; a role often played by the husband. Therefore, her movement was further restricted in his absence.

*"I cannot go to PHC alone, it's far from my village, so I have to ask other women [as husband is a migrant] to come with me, but everyone has their own work, who would come with me?"*—Woman with migrant husband, Age 28, FGD 6, High male-out migration area.

Sometimes stigma was attached to procuring of contraceptives from local pharmacies. Women who wanted to buy contraceptives from local pharmacies or shops could not do so due to the anticipated stigma attached to procuring of contraceptive methods. Fear of being judged as a promiscuous woman (who uses contraceptives in husband's absence) by the pharmacists/shop keeper restricted women from procuring contraceptive methods from private stores. Some women were apprehensive about family members and neighbours finding out about their contraceptive use if they procured contraceptives when husbands were not at home. Women with migrant husbands felt hesitant to discuss FP issues with ASHAs during husband's absence, fearing that their families may become suspicious. Women were anxious that their families might label them as *bad women* if they found them conversing on FP issues with ASHAs.

*"How can we go to medical stores and ask for contraceptives? Storekeepers might think, what will we do with contraceptives when husband is away?*—Woman with migrant husband, Age 25, FGD 3, High male-out migration area.

*"If husbands find out that we are using contraceptives when they are away working, they might suspect us of having affair with other men. That's why we do not use any method when they are away."*- Woman with migrant husband, Age 29, FGD 7, High male-out migration area.

**Pregnancy as a contraceptive.** Migrant men remained away from home for most part of the year. Some men who migrate internationally usually come home once in a year. The suspicion that wives might engage in extra-marital affairs when they are away existed among most married migrant men. Men felt that if their wife was pregnant, she would not engage in any extramarital sexual relationship during their absence. Driven by this suspicion men refrained from using any contraceptive when they were at home.

*"Some men in the village do not use any contraceptive methods to make sure their wife gets pregnant each time they visit home. If wife is pregnant while husband is away there is no chance of having extramarital relationships with other man."*–Married Migrant man, Age 29, FGD 1, High male-out migration area.

*"Every time migrant men visit home, their wives become pregnant, they deliberately do so to prevent wives from having affairs with other men behind their back".*—ASHA, Age 37, FGD 5, High male-out migration area.

### Reasons pertaining to migration environment

There were several migration environment related factors that acted as barrier to contraceptive use. These included supply side impediments and deep-rooted community norms.

**Low ASHA outreach on family planning.**   ASHAs are considered as key persons from the health system in the FP supply chain as they directly reach out to women at the village level. ASHAs were the only influencers and providers of contraceptives in the villages. However, they perceived migrant women were not conventional users of contraceptives. Outreach by ASHAs in villages with high male out-migration was poor. Almost all women and men across the discussions reported that ASHAs made limited number of visits and such visits were mainly for the purpose of child immunization and/or accompanying pregnant women for institutional delivery. Their limited reach to contraceptive users in high out-migration areas reduced women's timely access to contraceptives. Contraceptive use is conditioned upon decisions of both husband and wife. Therefore, it is important to counsel both men and women regarding the use of contraceptives. However, ASHAs did not talk to men in the villages about contraceptive use.

> "*ASHA sometimes comes to our village, but she only talks about immunization. She does not talk to me on contraceptives, maybe she thinks why to talk to us on contraceptive use since my husband is a migrant. She never talks to my husband [on FP] even when he visits home*"— Woman with migrant husband, Age 28, FGD 5, High male-out migration area.

The discussion with ASHAs on the issues of contraceptive was also limited as women perceived ASHAs would not keep their conversations confidential, especially if they discussed contraceptive issues when their husbands were away for work. ASHAs belonged to the same village and had close ties with most of the women, which made discussions around contraceptive use uncomfortable for women.

> "*We do not feel comfortable to talk to ASHAs on contraceptive use as they are native women; what if they tell everything we ask [when husbands were away] to our family and to other women in the community.*"—Woman with migrant husband, Age 27, FGD 7, High male-out migration area.

**Preference of male child.**   Preference of a male child was another important factor for non-use of contraceptives. Migrant families do not possess land or any other resources in their village. Their desire for male child is driven by the economic reasons. Many prefer a male child as it would mean more hands available to earn for the family.

Other reasons for preferring a male child are widely recognised in rural settings. Both women and men considered having a male child as a matter of family pride, since only sons could carry forward the family name. They also viewed having a male child was as support in their old age, as daughters would move to their husband's house after marriage. As a result, people did not want to use any contraceptive methods until they had a son.

> "*We are poor people, and we do not have any land, so if we have a son, he can go out like us and earn for the family.*"—Migrant man, Age 40, FGD 8, High male-out migration area.

> "*Even after having five daughters, the preference for a son is so high that they [couples] do not use anything [contraceptive method] and keep on having more children till they get a son.*"— ASHA, Age 40, FGD 4, High male-out migration area.

**Need to prove fertility.** Majority of young married couples did not use contraceptives owing to social pressure of bearing children soon after getting married, or else being labelled as infertile. This pressure is heightened among migrant families as men stay at home for a shorter period of time during the year and these couples get limited window of opportunity for reproduction. The unique issue of contraceptive non-use among newlywed couples is well established in the social setting. Even ASHAs refrained from visiting newly married women anticipating their mothers-in-law would not allow any talk on FP before the first pregnancy.

*"No one uses any method until the first birth, if we do not have a child soon after marriage, people might think us infertile."*- Woman with migrant husband, Age 18, FGD 3, High male-out migration area.

*"We do not visit the houses of newly-weds for FP related issues; they would not listen to us before they have their first child. Moreover, we do not want to get into fights with the mother-in-law for that [talking on FP]."*- ASHA, Age 45, FGD 4, High male-out migration area.

**Myths and misconceptions about contraception.** Limited knowledge about contraceptive methods resulted into myths and misconceptions which ultimately led to non-use of contraceptives by many women. Some of the myths and misconceptions about using contraceptive methods included infertility, long term health consequences, loss of stamina, damage of womb, cancer and even fear of death.

"*Women are afraid these [spacing] methods will harm the uterus and make future births risky.*"—ASHA, Age 31, FGD 4, High male-out migration area.

*"There are so many women who do not want to use any contraceptive methods, they are worried about side effects. This does not necessarily mean that they have used and experienced personally, most of the side effects are experienced by someone else."*—ASHA, Age 36, FGD 3, High male-out migration area.

*"In our neighbourhood one women died few days after getting operated [tubectomy], now other women are so frightened that no one in the village wants to go for the same"*.—Woman with migrant husband, FGD 3, High male-out migration area.

**Poor couple communication on contraceptive use.** Poor couple communication around contraceptive use was evident across all interviews and among all couples. The discussions showed the communication on FP issues took a backseat in cases where husbands were migrants. Couples who stayed apart had limited talks over phone and mostly conversed on household needs, children and other family issues. Both migrant men and women felt that contraceptive use was not relevant to discuss over phone. Further, when husbands visited home during their break, FP remained a distant topic of discussion. Women did not want to bring up FP issues fearing they may antagonise men and would bring marital conflicts. Women rather preferred to act obedient and followed what men asked them to do.

*"We do not bring up FP issues when husbands are at home, this might bring unnecessary conflict. We only do what they ask us to do."*—Woman with migrant husband, Age 20, FGD 8, High male-out migration area.

"*We don't talk to husband on these [FP] issues on phone, we mainly discuss household stuff.*"- Woman with migrant husband, Age, 28, FGD 7, High male-out migration area.

## Discussion

This study examined the reasons for non-use of contraceptives in the context of male out-migration. The barriers to contraceptive use are multifaceted, complex and reflect contraceptive use experiences of women and their peers. Reasons pertaining to both husband's migration and migration environment worked alone or in conjunction with the other factors that dissuaded women from using contraceptives. At the individual level, low perceived need for modern FP methods, lack of preparedness for contraceptive use during husband's visit, fear of side effects, inability to procure contraceptives, pregnancy perceived as a contraceptive were all reasons identified as barriers to contraceptive use. Whereas, at the system and community level, low ASHA outreach on FP, need to prove fertility, preference of male child, myths and misconceptions about contraception, poor couple communication on FP were listed as deterrents to contraceptive use. The reasons for contraceptive non-use identified in the study pointed out that migration as a context is important to explain the FP behaviours of couples, particularly in communities with high male out-migration. This is also supplemented by the recent evidence published by the study team which documented a lower rate of modern contraceptive use among women in areas with high volume of male out-migration than those in areas with low volume of out-migration [22]. The study further gives an indication on how some of the commonly believed reasons for non-use may have different meaning in the context of migration. While reasons for non-use among women with migrant husbands reflected their own experiences as well as experiences of other women in the community, the reasons identified in high out-migration area reflected community norms and supply side factors.

While multiple reasons contribute to the non-use of contraceptives in a migration environment, three reasons stood out. First is the poor outreach of ASHAs, second is preference for a male child and third is poor couple communication. ASHAs are the central pillar of India's FP program. Poor outreach by ASHAs, specifically in high out-migration areas is concerning. As the study findings suggested, ASHAs did not perceive the need for FP among women having migrant husbands and as a result the regular visits to homes of women did not take place. Notably, not only the women having migrant husbands were neglected, but even those having resident husbands were neglected in this process. This was an important supply side barrier to the non-use of contraceptives in the study area which might manifest individuals' resistant to contraceptive use hence to be addressed by the programmers and policy makers. The other factor that was widely discussed was the preference of male child. Study participants, specifically men, did not feel the need to use modern contraceptives unless they had the desired number of sons in the family [23–25]. Family size and decisions about use of contraceptives often depend upon the sex of the first child, with contraception being less likely until the birth of a male child [5,24]. The preference of male children among migrant families also had an added economic connotation attached to it, as male children were viewed as future income guarantee for the family. Male children unlike females could migrate to other areas and become the economic pillar for the family. Similar findings were also identified in other Indian studies as well [5,24].

Another factor that amplified the influence of other factors was poor couple communication. For example, poor spousal communication and limited conversations around FP could result in low perceived need of contraceptive use and low preparedness with procurement of contraceptives during husband's home visit. Both women with migrant husbands and migrant men mentioned that they did not discuss contraceptive use as it was not considered as important and relevant to discuss over limited conversations during telephone calls. Prior research examining poor couple communication on contraceptive use in India and elsewhere suggested that talks on reproduction were limited as conversations on sex and contraceptives were

considered as taboo [5,26,27]. Blanc et al (2001) suggest that gender-based power inequities contribute to poor spousal communication [28]. The imbalanced distribution of power between couples and the added societal pressure to prove fertility, specifically among young couples, jeopardizes women's intention to use contraceptive.

Among the individual factors, the inability to procure contraceptive methods reflected the accessibility to FP services in rural Bihar. Women in the discussion mentioned that they could not access contraceptive methods as they had no one to escort them to the facility in absence of their husband. In rural India, women need permission from family members for their movement to outside the house. The NFHS-4 study suggested that only 8% of women in Bihar could move without any restriction [4]. Restriction in mobility negatively affected women's health seeking behaviour [29]. Mobility restrictions could be more among women with migrant husbands as their companions were away for work. In addition to the restrictions, the stigma attached to procurement of contraceptives during husbands' absence was a key deterrent. Stigma as a barrier to women's access to several health services including sexual and reproductive health (SRH) has been documented in studies. These studies mentioned that several women in rural areas compromise in seeking SRH services due to their perceptions of how community members felt about such issues [30–33]. The other factors important from the individual perspective was low perceived need for contraceptive use and lack of preparedness with contraceptive method and among women with migrant husband Women with migrant husbands not perceiving the need to use contraceptive when the husband was away for work seemed to be natural. However, this likely led to a situation where women did not get sufficient time to procure methods when their husbands visited them. Also, as a result of low perceived need, communication of women with ASHAs on FP issues remained a distant topic to discuss. The study also noted that some migrant men use wives' pregnancy to safeguard wives' chastity during their absence and hence, avoided using any contraceptives. This was particularly true in the case of long-distance migrants who suspected that their wives might engage in an extramarital affair during their long absence. These men thought that if their wife was pregnant in their absence, she will be busy with caring of the child and hence, would not be engaged in extramarital sex. While each of the reasons identified in this research have their own importance, it may be possible that they are inter-dependent and may be triggered by another set of factors. Synthesis of the study findings and review of existing evidence on reasons for contraceptive non-use suggested that some of the reasons may be inter-connected and one may be aggravating the other (Fig 1). For example, low outreach of ASHAs in high out-migration areas may have a cascading effect on multiple individual level reasons, like low perceived need for contraceptives, which could further jeopardize the preparedness among women to procure contraceptives before husbands' visit home. Further, since in villages ASHAs were the primary providers of contraceptives, their low outreach to women on FP issues, deprived several women of correct knowledge on modern contraceptives and the ways to access those methods. It may also be hypothesized that this can lead to myths and misconceptions among women. Empirical evidence suggested only 25% of women in Bihar were met by an ASHA in a period of three months; of these, only 16% of women were informed on FP [4]. This suggested that within the limited number of women reached by ASHAs, even with them the discussion on FP was missing. Women in this study also reported that ASHAs' visit was mostly related to addressing maternal and child health issues.

Myths and misconceptions around modern contraceptives may have instilled fear of using contraceptives among women with migrant husbands. Studies from India and other developing countries have also shown fear of side effects as an important impediment to contraceptive use [5,34–37]. Negative experiences of contraceptive use were often talked in neighbourhoods which affected the willingness of others in the community to try contraceptive methods,

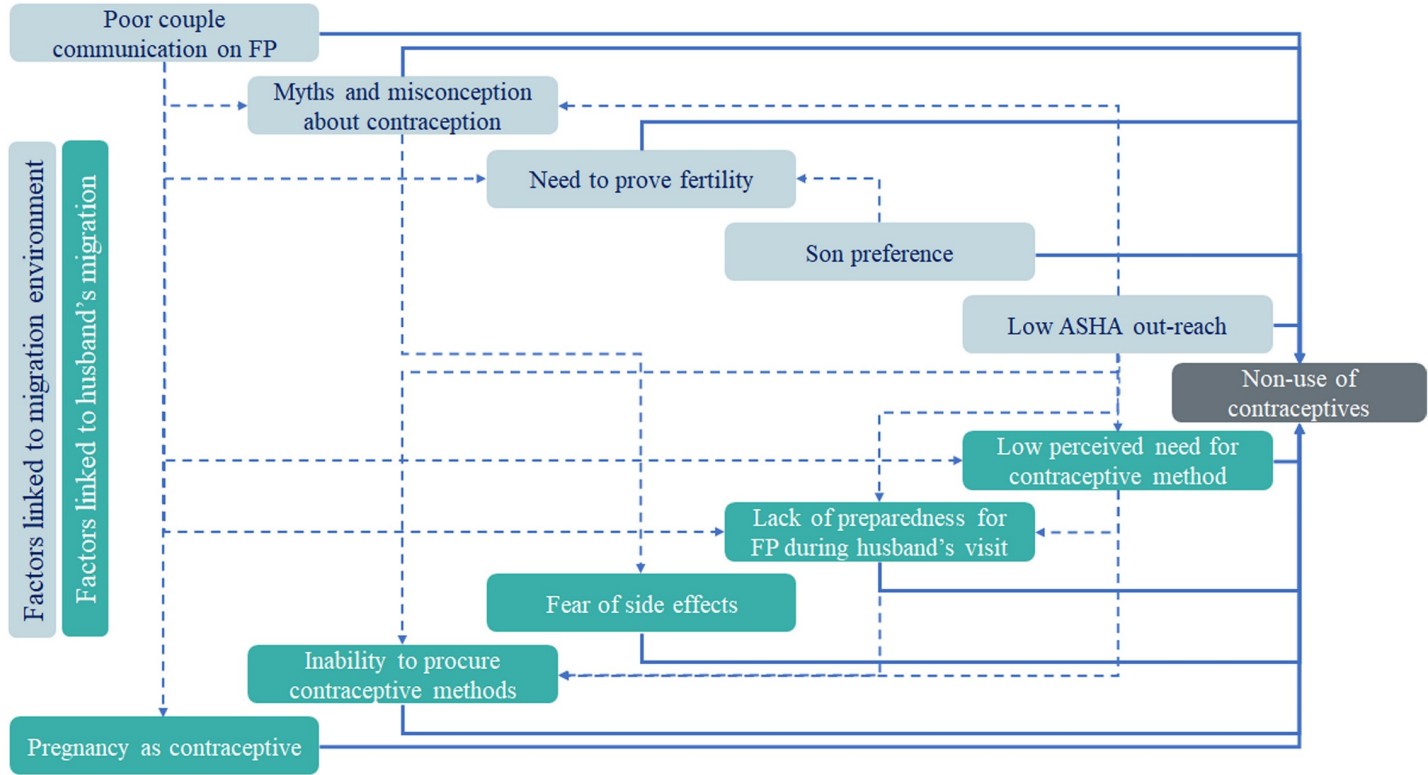

Note: Solid lines indicate direct linkage to contraceptive non-use and dashed lines indicate indirect pathways leading to contraceptive non-use

**Fig 1. Direct and indirect linkages between various factors leading to contraceptive non-use among women in Bihar.**

exacerbating fears of potential long-term negative health impacts [36,37]. During data collection, the study team also observed that there had been a couple of sterilization failure cases in the study area. As a result, several women preferred undergoing hysterectomy (assuming that is the safest way to stop bearing children). While myths and misconceptions around contraceptives are prevalent in the communities, the fear of sickness that stemmed from past use was another deterrent to contraceptive use. This study found that women with migrant husband did not use contraceptives due to their previous experience of falling sick (heavy bleeding, stomach pain) after using injectables or IUD. This was in line with prior studies which have also documented women discontinuing contraceptive methods as they fall sick once they start using contraceptive methods such as implants, injectables, IUDs and, oral pills. Being a primary caretaker of their households, women refrain from using contraceptives assuming if they fall sick who will take care of their children and other family members.

The study also found that the views of migrant and resident men on contraceptive use was mostly similar. Both groups of men did not perceive the need to use contraceptive methods. Driven by the belief that only men should decide the numbers and timing of the children, both groups also did not discuss contraceptive use and FP issues with their wives. However, the intention behind specific reasons for non-use of contraceptives differed between resident and migrant men. For example, while both reported preference of a male child, the intention for that was distinctly different between the two groups. The underlying reason behind resident men's preference for a male child was to carry forward family lineage or just a matter of pride in the community. However, migrant men preferred to have a male child for economic benefits. Since most migrant men were landless and belonged to the disadvantaged socio-economic

stratum of the society, they thought that male children unlike female children, could being economic stability to the families by out migrating like them. Given the differing migration patterns in the two study districts, the study also explored if the reasons for non-use differed between the study districts. However, not many w differences were found in respondent's reporting of reasons for not using contraceptives.

The study highlighted some pertinent reasons related to non-use of contraceptives. However, the study findings need to be treated cautiously to avoid over generalization due to some limitations. First, the study conducted interviews with women with migrant husbands, married men (both migrant and resident) and ASHAs. There could have been other respondents such as women with resident husband and female migrants. While interviewing them could have provided a wider understanding of contraceptive non-use, there is already ample information available on women with resident husbands. Even this study reflected some of the reasons (such as preference of a male child, myths and misconceptions, low ASHA outreach) identified in earlier studies. Second, the study did not conduct any in-depth interviews which would have helped in more in-depth understanding of each identified reason. Participants of the FGDs may not have talked openly in the group to certain sensitive topics. It is recommend that future research exploring similar issues should use a combination of FGDs and in-depth interviews for better exploration. The study recommends that future research studies should examine the root causes which may help FP programs in developing solutions to address those reasons.

Despite these limitations, this study provided important pointers for FP programs in Bihar and other areas affected by migration. Given that the reasons for non-use among women with migrant husbands are distinct from other women, FP programs need to devise appropriate strategies to address the gaps in existing programs. In addition, the reasons identified in the study in context of the migration environment provided a platform to work more effectively towards contraceptive non-use. Therefore, FP programs in areas affected by migration need to be more innovative and strategic in addressing barriers to contraceptive use. The reasons for non-use such as need to prove fertility, preference of male child and stigma attached to procuring contraception emerge from deep-seated social and cultural norms, which cannot be uprooted overnight. It is imperative to strategize effectively to work with communities to influence social and cultural norms. Findings of non-use around poor couple communication, and fear of side-effects reinforce the need for male involvement in contraceptive use, and FP improvement in couple dynamics and increase in knowledge of contraception methods. Moreover, efforts to strengthen community health workers to regularly visit and counsel on contraceptive use will improve FP communications in the community, encourage them to use contraceptives and help couples prepare themselves for the use of contraceptive methods. All such efforts would require sincere and shared responsibility of all FP gatekeepers—individuals, community and health system—to act together to build a conducive FP environment for men and women living in high out-migration areas.

## Conclusion

In conclusion, the study conducted in two high male out-migration areas of Bihar suggested that reasons for contraceptive non-use are multifaceted and inter-linked. The research revealed in-depth understanding of cultural, social and interpersonal contexts of contraceptive use behaviour in the context of migration. Programs working towards improving contraceptive use need to address issues at individual, community and system levels. While supply side factors like ASHA outreach need to be strengthened, prevailing social norms such as need to prove fertility, preference of male child, poor couple communication on contraceptives in the

community also need to change. Finally, the migration environment needs to be considered while designing FP programs to address special needs of women and couples living in the high out-migration setting.

## Author Contributions

**Conceptualization:** Saradiya Mukherjee, Bidhubhusan Mahapatra, Niranjan Saggurti.

**Data curation:** Saradiya Mukherjee.

**Formal analysis:** Saradiya Mukherjee.

**Funding acquisition:** Bidhubhusan Mahapatra, Niranjan Saggurti.

**Investigation:** Saradiya Mukherjee, Bidhubhusan Mahapatra.

**Methodology:** Bidhubhusan Mahapatra, Niranjan Saggurti.

**Project administration:** Bidhubhusan Mahapatra, Niranjan Saggurti.

**Supervision:** Niranjan Saggurti.

**Visualization:** Bidhubhusan Mahapatra.

**Writing – original draft:** Saradiya Mukherjee.

**Writing – review & editing:** Bidhubhusan Mahapatra, Niranjan Saggurti.

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
