## [Decision Letter · Decision Letter 0]

7 Sep 2020

PONE-D-20-13937

Why women do not use contraceptives: Exploring the role of male out-migration

PLOS ONE

Dear Dr. Mahapatra,

Thank you for submitting your manuscript to PLOS ONE. After careful consideration, we feel that it has merit but does not fully meet PLOS ONE’s publication criteria as it currently stands. Therefore, we invite you to submit a revised version of the manuscript that addresses the points raised during the review process.

Please closely review the comments from reviewers 1-3, particularly 1 and 3, regarding the expansion of the results and discussion.  Additionally, both the editor and Reviewer 1 indicated that more thought and justification is needed regarding the conflation of reasons related to migration and reasons that are not (e.g. experience of side-effects). 

The comments from the editor, in terms of checklists and additional language around disclosure statements, must be addressed. 

We look forward to receiving your revised manuscript.

Kind regards,

Linnea A Zimmerman, Ph.D, MPH

Academic Editor

PLOS ONE

Journal Requirements:

2.We note that you have indicated that data from this study are available upon request. PLOS only allows data to be available upon request if there are legal or ethical restrictions on sharing data publicly. For more information on unacceptable data access restrictions, please see http://journals.plos.org/plosone/s/data-availability#loc-unacceptable-data-access-restrictions.

Additional Editor Comments (if provided):

Because the research study is qualitative, the authors need to reframe their paper to ensure that it aligns with qualitative reporting guidelines. From the author guidelines: "Qualitative research studies should be reported in accordance to the Consolidated criteria for reporting qualitative research (COREQ) checklist or Standards for reporting qualitative research (SRQR) checklist." I suggest the COREQ checklist. Please ensure that the revised manuscript meets these guidelines and submit the guidelines, indicating where approximate revisions were made.

Many of the macro and micro influences that are named are non-specific to migration. There needs to be greater justification about why you consider these to be related to migration in both the introduction and discussion or separate these out as a separate category that is NOT specific to migration.

The results section needs considerable revision. Rather than presenting multiple quotes without much context, more work should be done to synthesize the themes, provide some general interpretation in terms of patterns within the themes and then provide 1-2 quotes as examples.

Data availability - more justification is needed on why data are unavailable.

Funding - Please ensure that the funding disclosure aligns with journal requirements (https://journals.plos.org/plosone/s/disclosure-of-funding-sources). The draft indicates that the work was supported by Packard, but the funding disclosure says that there was no specific funding. Please clarify.

Reviewers' comments:

Reviewer's Responses to Questions

**Comments to the Author**

1. Is the manuscript technically sound, and do the data support the conclusions?

Reviewer #1: Yes

Reviewer #2: No

Reviewer #3: Partly

Reviewer #4: Yes

2. Has the statistical analysis been performed appropriately and rigorously? 

Reviewer #1: N/A

Reviewer #2: N/A

Reviewer #3: Yes

Reviewer #4: N/A

3. Have the authors made all data underlying the findings in their manuscript fully available?

Reviewer #1: Yes

Reviewer #2: Yes

Reviewer #3: No

Reviewer #4: Yes

4. Is the manuscript presented in an intelligible fashion and written in standard English?

Reviewer #1: No

Reviewer #2: Yes

Reviewer #3: Yes

Reviewer #4: Yes

5. Review Comments to the Author

Reviewer #1: PONE-D-20-13937

Thank you for the opportunity to read this important and interesting paper. There are many exciting findings, which I think need to be published and with some work this paper should be published. Some of the findings are things that people suspect and talk about, but there have not been good studies actually documenting, so this is an important contribution. One of the most critical findings is the role of ASHAs in actually discouraging women from taking up a method when their husband is away! This has important programmatic implications. Also, the section on not wanting to use family planning before the first birth (for newly weds) is really important and helpful, I would cite this paper right now in another paper if I could! I have some larger comments here, and then more detailed comments about each specific section below.

Larger

1. Overall, your results need a little more interpretation/discussion. Right now its mostly a list of quotes with very little in between, but I think a little more context, helping us understand why you choose each quote, making sure that its more than a list of quotes is necessary. Please add some more discussion and contextualization to this section.

2. I do not quite understand why you have two separate sections, one about “fear of sickness” and one about “myths and misconceptions”—I think that these are probably the same topic, that there are perceptions of side effects, and I understand that the first set specifically mention migration, but I think people are talking about the same thing and that it would make sense to combine into one section. Also, since that first section just has people talking about “fear of sickness” with out us really knowing what they mean by that it is not very rich, so I would recommend just using 1 of the quotes in that section and merging with the other section which is richer.

3. The discussion and conclusion need the most work. Related to my point below (about knowing more about previous studies/findings), try to focus your discussion and conclusion especially on the new findings in your study, because I think there are important new findings and have big programmatic and policy implication, and that add to the literature, but right now they are lost in findings that we already know and that others have discussed. You can discuss these past findings and highlight what’s different for migrating couples, but I urge you to really reorganize and think through what your value add is, and highlight that.

4. The authors need to do another review of the literature, especially for the discussion, as many of their findings have been found by other authors (including myself, sorry for the self promotion). But look at these papers, and the papers they cite, and also do a literature search for others

a. https://pubmed.ncbi.nlm.nih.gov/22390371/

b. https://pubmed.ncbi.nlm.nih.gov/18821352/

5. A major limitation which needs to be discussed is that the men were not migrants, and in fact you were probably missing the migrant men since they were away. So this is really about the experiences of women with migrant husbands, and the perceptions/views of men and ASHA’s living in the same community. It's a difference between lived experiences and perceptions. Make this clear and discuss how this biases your results.

a.

6. Finally, this needs editing by a native English speaker. I have pointed out some things in the beginning, but common issues like this are pervasive throughout the paper.

7.

Watch switch between family planning and contraceptives—be consistent throughout your manuscript (example in the first paragraph)

Ln 49: “..and majority of..” should be “..and the majority of..”

Ln 65: “A recent research in Nepal..” should be “A recent study in Nepal..”

Ln 67: “Some of these studies..” should be cited

Ln 79: “moderated” sounds like you are doing a quantitative study, please use another word

Ln 80: “shape out” is not scientific and also not clear, please use a more scientific term

Ln 93: Identical? That seems like a very strong word to use, I doubt the are “identical”, replace with “very similar” , your next sentence also shows that there are differences

Ln 99: remove “the” before “qualitative data collection…”

Selection

• Was the recruitment done before the FGD and then the FGD arranged for a later time? What I am asking is if the respondents did the consent in a private location and then met later for the FGD or if the consenting was done at the time of the FGD? Please clarify in the text

• Where did the FGDs with men and women take place? How was privacy and confidentiality ensured (within the group members?)

• Couples were not linked right? (husband and wife both interviewed?)

• When were the ASHA’s consented? How did this process work? Add to the text. How did you ensure that they did not feel pressured to participate give the recruitment approach through their groups?

• Info about the consent should be earlier ( what is in the section about ethical approval should be moved up).

Data analysis

• How many people participated in the coding and analysis process?

• How did you deal with the fact that there are FGDs from three different groups? Were they analyzed by respondent type or all together? More detail is needed.

Results

• Ln 170: are these actual quote selections or your own interpretation? Make it clear.

• Might be useful to add respondents age and educational status too to the info after each quote

• Ln 172: “distorted” is a judgmental and charged word, also, it is incorporating analysis into the results, please rewrite this section

• Make sure not to sure contractions (such as “didn’t”) in formal academic writing

• Ln 205: I do not think you have defined PHC previously in the paper

• Any idea what women mean by “fall sick”, did you get any details about what sickness they are referring to?

• Ln 251: “They expressed if their wife was pregnant, they would not engage in any extramarital sexual relationship during their absence.” I had to read this a few times to figure out what you are trying to say, since its unclear who the various “they”’s are. I suggest re-writing as “Men felt that if their wife was pregnant, she would not engage in any extramarital sexual relationship during their absence.”

• Quote on ln 253: is this saying that they want their wife to become pregnant when they are home? I think you need some clarification/translation here otherwise it sounds like they want her to be pregnant while he is home, but I think its that they want her to be pregnant while he is away?

• Ln 261 “There were several macro-level factors, either directly or indirectly—through the micro level factors, acted as barrier to contraceptive use.” Re-write as ““There were several macro-level factors that acted as barrier to contraceptive use, either directly or indirectly through micro level factors. ”

• Can remove quote in line 315.

• Also, I would recommend removing the next two quotes and just saying “Respondents noted fears of infertility and cancer as reasons for non-use of family planning”

• Ln 329: I don't think that your first quote here is reflective of what you say in this paragraph about men thinking FP is “woman’s business” –this seems like he is saying that the husband decides about the number of children because he earns the money.

• Ln 348: I think you mean antagonize not “agonise”

Discussion

• In general, your discussion needs to be more specific and better situated in the literature. Especially for the part where you make recommendations, look for other studies of interventions or program/policies that might have worked to address your specific issues, even from other countries.

• You say that “aspiration to have smaller families” was universal, however, you have not mentioned that at all in the results, so you either need to mention that above or not mention it. Also, some of the quotes talked about people continuing to have children until a son, so it didn't seem that this was universal.

• Ln 370: I don't think your findings show anything about differences between men and women, or between migrants and non-migrants, since you only interviewed migrant wives. So I think this should be rephrased to only focus on your findings related to reasons for FP non-use/barriers to use among couples living in high migration areas

o This relates to the next sentence too—don’t extrapolate to non-migrants. I think many of your findings are similar to non-migrants (son preference, fear of side effects, proving fertility, fear of talk/social stigma, poor communication etc.). You could say that barriers that exist for other couples/women might be heightened among the migrant population.

• Ln 402: I really don't think that lack of communication is leading to son preference (debatable about the other too), and I am not sure about your point here. You have not shown evidence that couples did not understand each others preferences and therefore had more births than desired. Please be careful about your interpretation and making sure its reflective of your findings.

• Paragraph starting line 424: you present results in this paragraph, these should go in the results section. Or you should take out this part, its rather confusing, but maybe would be less confusing if you had presented these findings earlier. I am also still struggling with the “fear of sickness” being separate from “fears of side effects”

• Limitations: A main limitation is that the men were not all migrants, in fact, migrant men were probably less likely to be in the study because they were away, right? This is something that needs to be discussed in detail. Also, I don't think that the couples were linked, right? This is a limitation and would have given you much richer data.

• Ln 451: the reasons for non use are NOT distinct between women whose husbands migrate and those that do not. There are some different reasons, but many of the reasons are simply magnified by migration.

• Ln 453: The sentence starting “In addition..” is not clear, I do not know what you are trying to say.

Conclusion

• Ln 476: “prevailing fertility norms in the community also need to change” this is not a very specific and helpful comment, can you be more specific about what you mean and how this can happen?

Reviewer #2: This study aims to document the contraceptive use among women in a region of high male out-migration. This is a very interesting question to help discover new means to improve contraceptive uptake in this region. Nonetheless, I found that the methodology adopted to respond to the question is not adequate. First, contraceptive use is an intimate reality that many women will not want to discuss in group. While using focus group approach to discuss this sensitive event, the research question focus should not be on the individual but on the context. The results are presented as individual response to the research question. Therefore, it is not clear why the authors used focus group discussion for their study instead of individual in-depth interview.

The results presented are related to the specific context study (what the authors call macro) or to the husband’s migration (micro) whereas they are situation that is found in many other places. For instance, the “inability to procure contraceptive methods” have nothing to do with the fact that the husband is away or because of the migration area. This is a fact that many women lack the opportunity to get their methods in pharmacies or store. At the end, very few results are related to the specific context studied.

Finally, this study lacks to present the context of the study. How long do husbands in the village reside outside? Is it a long-term migration or a circular one? The experience of each woman is not similar even if they lived in a place of high out-migration. As the authors said before, an in-depth interview is more appropriate to understand women behaviour than the Focus Group discussion on individual and sensitive topics.

Reviewer #3: Thank you for the opportunity to review this interesting, thoughtful qualitative analysis of reasons for non-use of contraception among populations with high out-migration of males in Bihar, India. Please see my comments below for suggested revisions.

ABSTRACT

Line 29-30: This sentence implies that reasons for contraceptive non-use differed among women who had migrant husbands vs. resident husbands, and women who lived in high migration environments vs. those who did not. This does not reflect my understanding of the findings of the paper, as the methods section indicated that sampling was done solely in high-migration village clusters (so everyone lived in a high migration environment), and the results did not differentiate findings for migrant vs. resident husbands. Perhaps this could be rephrased to something like “The reasons for contraceptive non-use in areas with high male out-migration were complex, including reasons unique to high-migration settings and reasons commonly found in other settings.”

INTRODUCTION

-It might be helpful to offer a brief definition of “family planning” up front – i.e. a set of purposeful behaviors (timing and spacing/limiting of births through contraceptive use -- modern or traditional) undertaken to achieve desired family size.

Lines 54-55: “Even among those who used modern contraceptives, female sterilization was the most dominant method” – Can the authors elaborate briefly on why this is problematic? (i.e. a healthy contraceptive method mix should consist of a balance of short-acting, long-acting and permanent contraceptive methods. Skewed method mixes, particularly in younger populations, typically indicate issues with contraceptive services and/or uptake).

Lines 64-65: “male out-migration is associated with utilization of reproductive health services and contraceptive use among women in developing countries”. Suggest to say “negatively associated” to make it clear that the association is negative.

METHODS

Please add a brief paragraph on data collection specifying: Were the FGD guides structured, semi-structured, or unstructured? Who were the FGD facilitators (what were their qualifications, what training did they receive in FGD facilitation)? Was there a note-taker? Were FGDs recorded and transcribed, or was the analysis done using notes?

In the data analysis section (lines 147-153), please specify the following: was the coding done on full transcripts vs. notes? Was any double-coding done (two different people code the same transcript) to check inter-coder reliability?

Line 99: please correct to “focus group discussions”, not focused group discussions. Can the authors also briefly explain the rationale for selecting FGDs rather than in-depth interviews, given that discussing contraception could potentially be culturally sensitive in areas of Bihar with low contraceptive use rates? If FGDs were chosen for efficiency’s sake / budgetary reasons, that is fine. If the authors chose FGDs for additional reasons (e.g. they expected group discussions would elicit more interesting insights as respondents discussed experiences together), it would be helpful to specify this.

Lines 100-101: Can the authors briefly explain why they selected 2 blocks with high male out-migration and 2 blocks with low out-migration in each district? This is a bit confusing since the authors then say in lines 106-107 that they identified village clusters with high volume of high male out-migration. The results section does not distinguish between findings in high migration vs. low migration blocks, so it is unclear why this stratification was done.

Lines 108-110: Can the authors briefly explain the rationale for the sample size of 25 FGDs? For example, was this based on the expected likelihood of reaching saturation of themes after 8 or 9 FGDs per respondent type (women, men, ASHAs)? This would be helpful to clarify since line 112 says “the study team planned to conduct at least one FGD with each respondent group in each block” – which I take to mean 4 FGDs per respondent type (women / men), and the final count was twice that number (8 or 9 per respondent type).

RESULTS

Overall comment: Given the focus of this paper on contraceptive use in the context of high out-migration, I would suggest to restructure the results section to start with / emphasize the “reasons related to male out-migration” category. For the “reasons related to the migration environment” category, I would suggest to focus on reasons that are specific to a high out-migration setting (e.g. ASHAs tend not to do outreach since they don’t perceive a need for contraception when so many husbands are away; wives prioritize other issues to discuss / want to keep the peace during limited times when migrant husbands are home). I would suggest to cut back on the text describing other reasons for non-use that are already well documented in rural India and other settings (preference for male child, needing to prove fertility, myths/misperceptions about contraception). It should be sufficient to briefly describe these other reasons and state that even in a high out-migration environment, these common reasons persist as important barriers to use of contraception.

-There is little analysis of whether the results differed for the male FGD sub-types: married migrant men vs. married resident men. I would recommend to explore this further – did migrant men predominantly focus on their prolonged absence / lack of need for FP, vs. resident men focusing primarily on cultural/religious objections? If so, it would help nuance the paper’s main message – i.e. high out-migration settings experience same supply/demand challenges as other settings, but they also have unique challenges affecting all couples in the area (like ASHAs not doing outreach there) as well as challenges unique to migrant families (prolonged absence the common barriers to contraceptive uptake (supply/demand).

-It would be helpful to note whether/how the results differed in Gopalganj vs. Nawada. While their levels of out-migration are similar (approx. 1/3 of males), there is a considerable difference in contraceptive prevalence rate between these two districts (9% vs. 29%), so readers might be interested to know whether reasons for non-use differed.

Line 170-172: please clarify whether the words in italics are quotes from a respondent, or the authors’ interpretation. Also, the term “extremely distorted” seems pejorative -- I would suggest to replace this with a more neutral term like “founded in religious beliefs”.

Table 2: The terms “micro” and “macro” seem confusing here. These terms usually distinguish individual vs. higher-level factors (i.e. community/structural factors), and that’s not quite the case here. Both categories in Table 2 include a mix of individual, community, and structural factors. I would suggest deleting the terms “micro” and “macro”, and simply name these categories 1) Reasons pertaining to husband’s out-migration; 2) Other reasons (individual, community, structural).

DISCUSSION

In general, I would recommend being more cautious about the inferences drawn in the Discussion section. For example, lines 384-387 argue that the lack of ASHA outreach was responsible for several factors affecting contraceptive non-use (including low perceived need for contraceptives and myths/misperceptions about contraceptive methods), and lines 402-404 state that poor couple communication on FP may have led to son preference, need to prove fertility etc. -- but the Results section does not present data explicitly making these connections. In the absence of definitive evidence about these linkages in this study, I would recommend to either 1) cite the existing literature demonstrating these causal pathways; or 2) use more speculative language like “we hypothesize that lack of ASHA outreach may have contributed to…”

Lines 372-374: “The study suggests that reasons for contraceptive non-use when the husband was a migrant could be very different than when the husband was a resident and staying in a high out-migration environment.” I am not clear on how the study data supports this statement, since findings are not presented separately for migrant males vs. resident males, and the study did not interview women whose husbands were residents. Also, several reasons cited for non-use (cultural norms, misinformation, etc.) are common across other LMIC settings without high out-migration. I think it would be fair to say that the study demonstrates some reasons for contraceptive non-use that are unique to the high out-migration environment, and others that are common across multiple settings.

Figure 1: I appreciate that the authors are attempting to visually show the complex relationships among factors contributing to non-use of contraception. However, the indirect pathways shown in this figure are not fully supported by the data. For example, there is a dotted line (indirect link) from “poor couple communication” to “pregnancy as a contraceptive”, “need to prove fertility”, and “male child preference”, but the results section does not provide data indicating these factors are related in some way. If the authors decide to include this figure, I would suggest to clarify that the indirect pathways reflect the authors’ understanding of causal relationships/pathways in the existing literature, rather than empirical data from this study. Alternatively, I think this figure could be removed and the authors could simply state that the factors influencing contraceptive use in high-migration settings a) may differ in some ways from other settings, and b) are complex/nuanced (as is the case in most settings).

CONCLUSION

Line 472: add the word “contraceptive” before “non-use”.

Line 477-78: This final sentence feels a bit confusing. How would one put migration at the center of FP programs? Perhaps this could be reworded to say “the unique context of high out-migration settings needs to be considered and addressed in FP programs”

-See above comment about use of the terms “micro” and “macro” – this feels confusing, as these terms usually distinguish individual vs. community/structural factors, and that’s not quite the case here.

Reviewer #4: Methodology is sound, well-explained, thorough, with very good sample size. A couple of items where clarifications are needed: (1) The introduction comments on HIV as a problem among those not using any contraception, but results and conclusion do not discuss barrier methods vs other methods that would not protect against sexually-transmitted disease. (2) There is also not enough comment in the results or discussion on permanent or longer-term (tubal ligation, IUD) vs temporary contraception. Given inability to procure on time before husband's arrival is an issue, one would assume most of the discussions revolved around temporary contraception (pill, condom, etc), but there are mentions of tubal ligations and some fears of IUDs causing health issues. This needs further discussion or clarification.

6. PLOS authors have the option to publish the peer review history of their article (what does this mean?). If published, this will include your full peer review and any attached files.

Reviewer #1: **Yes: **Nadia Diamond-Smith

Reviewer #2: No

Reviewer #3: No

Reviewer #4: No

---

## [Author Response · Author response to Decision Letter 0]

18 Nov 2020

Manuscript title: Why women do not use contraceptives: Exploring the role of male out-migration

Manuscript ID: PONE-D-20-13937

Dear Dr. Zimmerman,

Academic Editor

PLOS ONE

Thank you for giving us an opportunity to revise our paper based on reviewers’ suggestions. We thank all the reviewers for their thoughtful comments, which undoubtedly have helped improve the quality of the manuscript. We have made the necessary changes to the manuscript. Following is our response to the reviewers' comments, as required. 

Journal requirements

1. If there are ethical or legal restrictions on sharing a de-identified data set, please explain them in detail (e.g., data contain potentially sensitive information, data are owned by a third-party organization, etc.) and who has imposed them (e.g., an ethics committee). Please also provide contact information for a data access committee, ethics committee, or other institutional body to which data requests may be sent.

Reply: Given the data comes from focus group discussions with women. At certain instances, the identifiers are part of the data itself. Therefore, it would be possible to de-identify them thoroughly. In this backdrop, the data used cannot be shared publicly. In case, someone wants to use the data, they write individually to Dr. Bidhubhusan Mahapatra, study principal investigator (bbmahapatra@popcouncil.org) to access the data. 

Response to the Editor

1. Because the research study is qualitative, the authors need to reframe their paper to ensure that it aligns with qualitative reporting guidelines. From the author guidelines: "Qualitative research studies should be reported in accordance to the Consolidated criteria for reporting qualitative research (COREQ) checklist or Standards for reporting qualitative research (SRQR) checklist." I suggest the COREQ checklist. Please ensure that the revised manuscript meets these guidelines and submit the guidelines, indicating where approximate revisions were made.

Reply: Thanks for the suggestion. We have gone through the COREQ checklist and made necessary modifications in the paper. 

2. Many of the macro and micro influences that are named are non-specific to migration. There needs to be greater justification about why you consider these to be related to migration in both the introduction and discussion or separate these out as a separate category that is NOT specific to migration.

Reply: Some of the reasons may sound non-specific to migration. But, when viewed from the context of migration environment, these reasons make more sense. For example, the son preference seems like to be generic reason for non-use. However, when one puts this in migration context, the son preference has a different meaning in areas affected by migration than not. In migration affected areas, the economic dependency on male child is distinctly different from low migration areas where son preference is more for lineage. In the revised manuscript, we have included justifications on these lines to make it more relevant. 

3.The results section needs considerable revision. Rather than presenting multiple quotes without much context, more work should be done to synthesize the themes, provide some general interpretation in terms of patterns within the themes and then provide 1-2 quotes as examples.

Reply: Thanks for the suggestion. We have made changes in the result section as per the suggestion. 

4. Data availability - more justification is needed on why data are unavailable.

Reply: Given the data comes from focus group discussions with women. At certain instances, the identifiers are part of the data itself. Therefore, it would be possible to de-identify them thoroughly. In this backdrop, the data used cannot be shared publicly. In case, someone wants to use the data, they write individually to Dr. Bidhubhusan Mahapatra, study principal investigator (bbmahapatra@popcouncil.org) to access the data.

5. Funding - Please ensure that the funding disclosure aligns with journal requirements (https://journals.plos.org/plosone/s/disclosure-of-funding-sources). The draft indicates that the work was supported by Packard, but the funding disclosure says that there was no specific funding. Please clarify.

Reply: The data collection was supported by Packard. However, for the paper analysis and publication, Packard has not provided any support. 

Response to the reviewer #1: 

1. Overall, your results need a little more interpretation/discussion. Right now, it’s mostly a list of quotes with very little in between, but I think a little more context, helping us understand why you choose each quote, making sure that its more than a list of quotes is necessary. Please add some more discussion and contextualization to this section.

Reply: Thank you for the feedback. As suggested, we have tried to contextualize the results to make it more understandable. 

2. I do not quite understand why you have two separate sections, one about “fear of sickness” and one about “myths and misconceptions”—I think that these are probably the same topic, that there are perceptions of side effects, and I understand that the first set specifically mention migration, but I think people are talking about the same thing and that it would make sense to combine into one section. Also, since that first section just has people talking about “fear of sickness” without us really knowing what they mean by that it is not very rich, so I would recommend just using 1 of the quotes in that section and merging with the other section which is richer.

Reply: We agree to the reviewer’s view to a certain extent that fear of sickness may be arising out of myths and misconceptions and this is something already depicted in the conceptual diagram as well. However, we disagree that they reflect the same issue. Fear of sickness is not only an outcome of myths and misconception but is a result of multiple factors. While fear of sickness is mostly based on women’s own experience, myths and misconception is a reflection of poor knowledge about contraceptive in the community. For example, our study found that women had used IUCD or injectables had experienced vaginally bleeding. As a result, they stopped using contraceptive going forward. This is fear of sickness based on their own experience. What we observed is that the sense of fear was particularly high among women having migrant husband, who thinks that if they experience something similar again, who will take care of them and their family in husband’s absence. 

3. The discussion and conclusion need the most work. Related to my point below (about knowing more about previous studies/findings), try to focus your discussion and conclusion especially on the new findings in your study, because I think there are important new findings and have big programmatic and policy implication, and that add to the literature, but right now they are lost in findings that we already know and that others have discussed. You can discuss these past findings and highlight what’s different for migrating couples, but I urge you to really reorganize and think through what your value add is and highlight that.

Reply: Thank you for the suggestion. As suggested, we have re-worked on the discussion and conclusion to highlight new findings that can have programmatic implications. 

4. The authors need to do another review of the literature, especially for the discussion, as many of their findings have been found by other authors (including myself, sorry for the self-promotion). But look at these papers, and the papers they cite, and also do a literature search for others: https://pubmed.ncbi.nlm.nih.gov/22390371/;
https://pubmed.ncbi.nlm.nih.gov/18821352/

Reply: Thank you for highlighting and sharing the link to very interesting papers. We re-examined the existing literature and included them as appropriate. 

5. A major limitation which needs to be discussed is that the men were not migrants, and in fact you were probably missing the migrant men since they were away. So this is really about the experiences of women with migrant husbands, and the perceptions/views of men and ASHA’s living in the same community. It's a difference between lived experiences and perceptions. Make this clear and discuss how this biases your results.

Reply: Thanks for the suggestions. The FGDs with men included both migrant (who were visiting the villages that time) and resident men. Since the study included both migrant and resident men, the perceptions of non-use of contraceptives were a mix of own experiences and viewpoints of resident men. However, the purpose of the study was to understand the reasons for contraceptive non-use in high male out-migration areas and how the perceptions on contraceptives influenced by virtue of living in such areas. Therefore, we also included resident men in the FGDs. Given this, we don’t think the results are biased from this perspective. 

6. Finally, this needs editing by a native English speaker. I have pointed out some things in the beginning, but common issues like this are pervasive throughout the paper.

-Watch switch between family planning and contraceptives—be consistent throughout your manuscript (example in the first paragraph)

- Ln 49: “..and majority of..” should be “..and the majority of..”

- Ln 65: “A recent research in Nepal..” should be “A recent study in Nepal..”

- Ln 67: “Some of these studies..” should be cited

Reply: We have carefully reviewed the language used in the paper and corrected wherever appropriate. 

7. Ln 79: “moderated” sounds like you are doing a quantitative study, please use another word

Reply: We have replaced the word “moderated” and used “influenced” in the text. 

8. Ln 80: “shape out” is not scientific and also not clear, please use a more scientific term

Reply: We have replaced the word “shape out” and used “could be viewed from the lens of migration” in the text.

9. Ln 93: Identical? That seems like a very strong word to use, I doubt the are “identical”, replace with “very similar”, your next sentence also shows that there are differences

Reply: We have replaced the word “identical” and used “similar” in the text

10.Ln 99: remove “the” before “qualitative data collection…”

Reply: We have removed “the” before “qualitative data collection” in the revised manuscript. 

11. Was the recruitment done before the FGD and then the FGD arranged for a later time? What I am asking is if the respondents did the consent in a private location and then met later for the FGD or if the consenting was done at the time of the FGD? Please clarify in the text

Reply: Yes, first oral consent was taken at a private location before individuals met together for the FGDs. 

12. Where did the FGDs with men and women take place? How was privacy and confidentiality ensured (within the group members?)

Reply: All the FGDs were conducted in a place where the discussion cannot be heard by outsiders. Women FGDs were conducted in women-only environment, men FGDs were conducted in a men-only environment. In villages, we mostly rented out village clubs (hall where village meetings are held) or temple premise to conduct these interviews. 

13. Couples were not linked right? (husband and wife both interviewed?)

Reply: Yes, couples were not linked. 

14. When were the ASHA’s consented? How did this process work? Add to the text. How did you ensure that they did not feel pressured to participate give the recruitment approach through their groups?

Reply: Like women, ASHAs also consented in two stages: first, verbally before coming to the interview; second, written after coming to the venue of FGD. As described in the manuscript, we selected 10 ASHAs from each PHCs approached them individually for their participation. ASHAs who provided verbal consent were again asked to provide written consent on the day of FGDs. There was no pressure on ASHAs to participate in the study as the research investigator clearly explained the voluntary nature of participation and information that will be sought from them. The absence of pressure can be also seen from the fact that some of the ASHAs actually refused to participate in the FGDs. 

15. Info about the consent should be earlier (what is in the section about ethical approval should be moved up).

Reply: In the revised manuscript, we have moved the text as suggested. 

16. How many people participated in the coding and analysis process?

Reply: Codes were developed by two researchers independently. Once the list of codes were prepared, researchers discussed amongst themselves and prepare a draft list of codes. The draft list was reviewed by the principal investigator of the study, who made final decisions on coded list.

17. How did you deal with the fact that there are FGDs from three different groups? Were they analyzed by respondent type or all together? More detail is needed.

Reply: All data from the interviews with women, men and ASHAs were coded and analysed separately. We have amended this in the revised manuscript.

18. Ln 170: are these actual quote selections or your own interpretation? Make it clear.

Reply: Thank you for highlighting this. They are our own interpretations. We have taken out the quotation marks to make it clear. 

19. Might be useful to add respondents age and educational status too to the info after each quote

Reply: We have added the age of the respondents after each quote. 

20. Ln 172: “distorted” is a judgmental and charged word, also, it is incorporating analysis into the results, please rewrite this section

Reply: Thank you for highlighting this. We have changed the word “distorted” to “not considered important” in the revised manuscript.

21. Make sure not to sure contractions (such as “didn’t”) in formal academic writing

Reply: Thank you. We have corrected these in the revised manuscript. 

22. Ln 205: I do not think you have defined PHC previously in the paper 

Reply: Yes, we already defined PHC in the method section. 

23. Any idea what women mean by “fall sick”, did you get any details about what sickness they are referring to? 

Reply: During the discussion women mentioned that they suffered from heavy vaginal bleeding after using injectables (called Antara injection in the study area). Some of the women also complained about bleeding due to IUD insertion.

24. Ln 251: “They expressed if their wife was pregnant, they would not engage in any extramarital sexual relationship during their absence.” I had to read this a few times to figure out what you are trying to say, since its unclear who the various “they”’s are. I suggest re-writing as “Men felt that if their wife was pregnant, she would not engage in any extramarital sexual relationship during their absence.”

Reply: Thanks for the suggestion. We have re-written as suggested. 

25. Quote on ln 253: is this saying that they want their wife to become pregnant when they are home? I think you need some clarification/translation here otherwise it sounds like they want her to be pregnant while he is home, but I think its that they want her to be pregnant while he is away?

Reply: Yes, men want their wives to be pregnant when they were away. We have updated the text as below. 

“Men felt that if their wife was pregnant, she would not engage in any extramarital sexual relationship during their absence. Driven by this suspicion men refrained from using any contraceptive when they were at home so that their wives become pregnant when they were away from home”.

26. Ln 261 “There were several macro-level factors, either directly or indirectly—through the micro level factors, acted as barrier to contraceptive use.” Re-write as ““There were several macro-level factors that acted as barrier to contraceptive use, either directly or indirectly through micro level factors.”

Reply: Suggested changes have been made in the text.

27. Can remove quote in line 315.

Reply: We have delated the quote in the revised manuscript.

28. Also, I would recommend removing the next two quotes and just saying “Respondents noted fears of infertility and cancer as reasons for non-use of family planning” 

Reply: We feel they are important quotes relevant to myths and misconceptions. Therefore, we have decided to keep those in the manuscript. 

29. Ln 329: I don't think that your first quote here is reflective of what you say in this paragraph about men thinking FP is “woman’s business” –this seems like he is saying that the husband decides about the number of children because he earns the money.

Reply: Thank you for highlighting this. We have removed the suggested quote. 

30. Ln 348: I think you mean antagonize not “agonise”.

Reply: Yes, we have updated that in the revised manuscript.

31. In general, your discussion needs to be more specific and better situated in the literature. Especially for the part where you make recommendations, look for other studies of interventions or program/policies that might have worked to address your specific issues, even from other countries.

Reply: As suggested, we have made changes in the discussion part. We would like to highlight that migration specific family planning interventions do not exist in India currently. We found some relevant ones from Nepal and our recommendations are keeping those in mind. 

32. You say that “aspiration to have smaller families” was universal, however, you have not mentioned that at all in the results, so you either need to mention that above or not mention it. Also, some of the quotes talked about people continuing to have children until a son, so it didn't seem that this was universal.

Reply: Yes, rightly pointed out, hence we have deleted the statement from the revised manuscript. 

33. Ln 370: I don't think your findings show anything about differences between men and women, or between migrants and non-migrants, since you only interviewed migrant wives. So I think this should be rephrased to only focus on your findings related to reasons for FP non-use/barriers to use among couples living in high migration areas

Reply: As suggested, we have re-phrased to the sentence to indicate the study shows reasons for non-use among couples in high outmigration settings.

34. The reasons for contraceptive non-use identified in the study pointed out that migration as a context is important to explain the FP behaviours of women and men, particularly, in communities with high male out-migration. This relates to the next sentence too—don’t extrapolate to non-migrants. I think many of your findings are similar to non-migrants (son preference, fear of side effects, proving fertility, fear of talk/social stigma, poor communication etc.). You could say that barriers that exist for other couples/women might be heightened among the migrant population.

Reply: We would like to disagree with the reviewer here. We are not extrapolating it to non-migrants. We are referring to the migration environment which included both non-migrant and migrant. It is not true that only women having migrant husband were part of the study, but also included both resident and migrant men as well as ASHA. Many of the environment related factors emerged from discussion with all three and hence, we feel the statement is relevant.

35. Ln 402: I really don't think that lack of communication is leading to son preference (debatable about the other too), and I am not sure about your point here. You have not shown evidence that couples did not understand each other’s preferences and therefore had more births than desired. Please be careful about your interpretation and making sure its reflective of your findings.

Reply: Your suggestion is much appreciated. We have removed the suggested part from the sentence. 

36. Paragraph starting line 424: you present results in this paragraph, these should go in the results section. Or you should take out this part, its rather confusing, but maybe would be less confusing if you had presented these findings earlier. I am also still struggling with the “fear of sickness” being separate from “fears of side effects”

Reply: Thank you for highlighting this. We have re-written the part of the sentence to bring clarity on this. We did not include it as part of result as they were observations by the study team and not from the respondents directly. 

37. Limitations: A main limitation is that the men were not all migrants, in fact, migrant men were probably less likely to be in the study because they were away, right? This is something that needs to be discussed in detail. Also, I don't think that the couples were linked, right? This is a limitation and would have given you much richer data.

Reply: We had both migrant and non-migrant men in the study and almost in equal proportions. Further, we don’t think having linked couples would have provided rich data. Given that there are FGDs, participation of couple would not be providing any additional information that what is presented in the study currently. 

38. Ln 451: the reasons for non use are NOT distinct between women whose husbands migrate and those that do not. There are some different reasons, but many of the reasons are simply magnified by migration.

Reply: We feel they are distinct, specifically, the individual specific reasons, Yes, the migration environment specific reasons are not distinct, but, heightened by the migration environment. We have re-written the part of bring clarity around this. 

39. Ln 453: The sentence starting “In addition..” is not clear, I do not know what you are trying to say.

Reply: We have modified the sentence for better understanding in the manuscript.

40. Ln 476: “prevailing fertility norms in the community also need to change” this is not a very specific and helpful comment, can you be more specific about what you mean and how this can happen?

Reply: We have changed “prevailing fertility norms” to prevailing social norms and updated the text in the manuscript. Prevailing social norms such as need to prove fertility, preference of male child, poor couple communication on contraceptives in the community also need to change. These norms could be changed through implementation of behaviour change intervention.

Response to the reviewer #2: 

1. This study aims to document the contraceptive use among women in a region of high male out-migration. This is a very interesting question to help discover new means to improve contraceptive uptake in this region. Nonetheless, I found that the methodology adopted to respond to the question is not adequate. First, contraceptive use is an intimate reality that many women will not want to discuss in group. While using focus group approach to discuss this sensitive event, the research question focus should not be on the individual but on the context. The results are presented as individual response to the research question. Therefore, it is not clear why the authors used focus group discussion for their study instead of individual in-depth interview.

Reply: While in-depth interviews could have provided finer nuances on reasons for not using contraceptive; however, it is not true that FGDs are not able to unravel the issue. We agree that the issue has to be discussed based on the context. Group discussions could elicit interesting insights as respondents discussed experiences together about the contraceptive use of their experiences and general practices about other women of the area. We disagree that results are presented as individual response. Yes, in some instances, during the discussions a few women mentioned their own experiences and we decided to keep these quotes while writing the results.

2.The results presented are related to the specific context study (what the authors call macro) or to the husband’s migration (micro) whereas they are situation that is found in many other places. For instance, the “inability to procure contraceptive methods” have nothing to do with the fact that the husband is away or because of the migration area. This is a fact that many women lack the opportunity to get their methods in pharmacies or store. At the end, very few results are related to the specific context studied.

Reply: While we agree some of the reasons identified looks generic in nature when looked from an overview perspective, the reasons have a different meaning when examined in migration specific context. If we take reviewer’s example on “inability to procure contraceptive methods”, yes, women lack opportunities to get their methods in pharmacies or store. But, it has different meaning for women having migrant husband. The mobility of women having migrant husband is limited when husband is away. She cannot go alone to a health centre to procure contraceptive. Moreover, since in villages procuring contraceptive is a sign of sexual activity, and women often fear that they would be judged as “promiscuous” if they procure contraceptives from shops or pharmacists while their husbands were away. 

3. Finally, this study lacks to present the context of the study. How long do husbands in the village reside outside? Is it a long-term migration or a circular one? 

Reply: We have already included the study context as part of the methodology. We have included a few more sentences to provide a better idea about the migration pattern in the revised version. 

4. The experience of each woman is not similar even if they lived in a place of high out-migration. As the authors said before, an in-depth interview is more appropriate to understand women behaviour than the Focus Group discussion on individual and sensitive topics.

Reply: We agree that the experiences can be diverging and that’s where FGD can bring those divergent experiences and still provides a coherent summary of the situation in the community. 

Response to the reviewer #3: 

1. Line 29-30: This sentence implies that reasons for contraceptive non-use differed among women who had migrant husbands vs. resident husbands, and women who lived in high migration environments vs. those who did not. This does not reflect my understanding of the findings of the paper, as the methods section indicated that sampling was done solely in high-migration village clusters (so everyone lived in a high migration environment), and the results did not differentiate findings for migrant vs. resident husbands. Perhaps this could be rephrased to something like “The reasons for contraceptive non-use in areas with high male out-migration were complex, including reasons unique to high-migration settings and reasons commonly found in other settings.”

Reply: Thank you for the suggestion. We made the suggested change in the revised manuscript. 

2.It might be helpful to offer a brief definition of “family planning” up front – i.e. a set of purposeful behaviors (timing and spacing/limiting of births through contraceptive use -- modern or traditional) undertaken to achieve desired family size.

Reply: We have included briefly on definition of family planning in the revised manuscript. 

3. Lines 54-55: “Even among those who used modern contraceptives, female sterilization was the most dominant method” – Can the authors elaborate briefly on why this is problematic? (i.e. a healthy contraceptive method mix should consist of a balance of short-acting, long-acting and permanent contraceptive methods. Skewed method mixes, particularly in younger populations, typically indicate issues with contraceptive services and/or uptake).

Reply: We have explained the problem with sterilization in the revised manuscript. 

4.Lines 64-65: “male out-migration is associated with utilization of reproductive health services and contraceptive use among women in developing countries”. Suggest saying “negatively associated” to make it clear that the association is negative.

Reply: Thanks for suggestion. We have updated in the revised manuscript.

5. Please add a brief paragraph on data collection specifying: Were the FGD guides structured, semi-structured, or unstructured? Who were the FGD facilitators (what were their qualifications, what training did they receive in FGD facilitation)? Was there a note-taker? Were FGDs recorded and transcribed, or was the analysis done using notes?

Reply: The details on interview procedure for conducting FGDs has been included in the revised manuscript in a separate heading called “interview procedure”.

6.In the data analysis section (lines 147-153), please specify the following: was the coding done on full transcripts vs. notes? Was any double-coding done (two different people code the same transcript) to check inter-coder reliability?

Reply: As suggested, we have included the details in the revised manuscript. 

7. Line 99: please correct to “focus group discussions”, not focused group discussions. Can the authors also briefly explain the rationale for selecting FGDs rather than in-depth interviews, given that discussing contraception could potentially be culturally sensitive in areas of Bihar with low contraceptive use rates? If FGDs were chosen for efficiency’s sake / budgetary reasons, that is fine. If the authors chose FGDs for additional reasons (e.g. they expected group discussions would elicit more interesting insights as respondents discussed experiences together), it would be helpful to specify this.

Reply: Thank you for this comment. We have included a couple sentence on the rationale behind going for FGDs. In summary, the FGD was preferred as they help in understanding a broader view of the situation in a community. Also, FGDs were done so that they can inform tool development for quantitative study. 

8. Lines 100-101: Can the authors briefly explain why they selected 2 blocks with high male out-migration and 2 blocks with low out-migration in each district? This is a bit confusing since the authors then say in lines 106-107 that they identified village clusters with high volume of high male out-migration. The results section does not distinguish between findings in high migration vs. low migration blocks, so it is unclear why this stratification was done.

Reply: When we started the study, we assumed that there may be some differences in pattern of contraceptive use between low and high migration blocks. However, at the analysis stage, we did not find such differences and hence, results were presented without any further segregation. However, such segregation helped us in picking the migration environment specific reasons and the finer nuances that differentiates the reasons of non-use between high out-migration and low out-migration communities.

9. Lines 108-110: Can the authors briefly explain the rationale for the sample size of 25 FGDs? For example, was this based on the expected likelihood of reaching saturation of themes after 8 or 9 FGDs per respondent type (women, men, ASHAs)? This would be helpful to clarify since line 112 says “the study team planned to conduct at least one FGD with each respondent group in each block” – which I take to mean 4 FGDs per respondent type (women / men), and the final count was twice that number (8 or 9 per respondent type). 

Reply: There is some confusion here. We conducted one FGD for each study group in each of the eight blocks. For men, we had to conduct an additional FGD in a block to capture the diverging views across population groups in that block. 

10. Overall comment: Given the focus of this paper on contraceptive use in the context of high out-migration, I would suggest to restructure the results section to start with / emphasize the “reasons related to male out-migration” category. For the “reasons related to the migration environment” category, I would suggest to focus on reasons that are specific to a high out-migration setting (e.g. ASHAs tend not to do outreach since they don’t perceive a need for contraception when so many husbands are away; wives prioritize other issues to discuss / want to keep the peace during limited times when migrant husbands are home). I would suggest to cut back on the text describing other reasons for non-use that are already well documented in rural India and other settings (preference for male child, needing to prove fertility, myths/misperceptions about contraception). It should be sufficient to briefly describe these other reasons and state that even in a high out-migration environment, these common reasons persist as important barriers to use of contraception.

Reply: In the original manuscript, the results are already presented first for “reasons related to male out-migration” and then for the “migration environment” related reasons which is line with the suggestion. On the second suggestion to cut back text for some of the migration environment related reasons, we have a different viewpoint. While some of the reasons may sound like already discussed in prior literature, it is important to present them in the migration context. Each of the reasons presented in the paper has their own relevance when discussed from migration perspectives. For example, son preference is very common in any rural setting in India. However, some preference in a non-migration setting has a different connotation than that in migration affected settings. In migration affected settings, men and women preferred son over a daughter due to economical reasons. In contrary, in a low out-migration setting, family lineage is the primary reason for son preference.

11. There is little analysis of whether the results differed for the male FGD sub-types: married migrant men vs. married resident men. I would recommend to explore this further – did migrant men predominantly focus on their prolonged absence / lack of need for FP, vs. resident men focusing primarily on cultural/religious objections? If so, it would help nuance the paper’s main message – i.e. high out-migration settings experience same supply/demand challenges as other settings, but they also have unique challenges affecting all couples in the area (like ASHAs not doing outreach there) as well as challenges unique to migrant families (prolonged absence the common barriers to contraceptive uptake (supply/demand). 

Reply: Thank you for suggesting this. In the revised manuscript, we have highlighted whether the results referred to migrant or resident men. Also, in the discussion, we have included briefly how the responses varied between resident and migrant men.

12. It would be helpful to note whether/how the results differed in Gopalganj vs. Nawada. While their levels of out-migration are similar (approx. 1/3 of males), there is a considerable difference in contraceptive prevalence rate between these two districts (9% vs. 29%), so readers might be interested to know whether reasons for non-use differed. 

Reply: Thanks for the suggestion. We have discussed the differences between Nawada and Gopalganj in the revised manuscript. 

13. Line 170-172: please clarify whether the words in italics are quotes from a respondent, or the authors’ interpretation. Also, the term “extremely distorted” seems pejorative -- I would suggest to replace this with a more neutral term like “founded in religious beliefs”.

Reply: Thank you for highlighting this. The words in italics were our interpretation. So, we have made the corrections in the revised version. 

14. Table 2: The terms “micro” and “macro” seem confusing here. These terms usually distinguish individual vs. higher-level factors (i.e. community/structural factors), and that’s not quite the case here. Both categories in Table 2 include a mix of individual, community, and structural factors. I would suggest deleting the terms “micro” and “macro”, and simply name these categories 1) Reasons pertaining to husband’s out-migration; 2) Other reasons (individual, community, structural).

Reply: Thanks for the suggestion. We understand that some of the factors included under macro are not actually at macro level. As suggested, we have removed the word macro and micro and have divided the reasons into two groups: reasons pertaining to husband’s out-migration and reasons pertaining to migration environment. 

15. In general, I would recommend being more cautious about the inferences drawn in the Discussion section. For example, lines 384-387 argue that the lack of ASHA outreach was responsible for several factors affecting contraceptive non-use (including low perceived need for contraceptives and myths/misperceptions about contraceptive methods), and lines 402-404 state that poor couple communication on FP may have led to son preference, need to prove fertility etc. -- but the Results section does not present data explicitly making these connections. In the absence of definitive evidence about these linkages in this study, I would recommend to either 1) cite the existing literature demonstrating these causal pathways; or 2) use more speculative language like “we hypothesize that lack of ASHA outreach may have contributed to…” “For lines 402-404 state that poor couple communication on FP may have led to son preference, need to prove fertility etc”.

Reply: Thanks for highlighting this. We have removed the sentence linking poor couple communication to son preference and need to prove fertility. Also, as suggested, we have re-articulated the language to make it look hypothesized. 

16. Lines 372-374: “The study suggests that reasons for contraceptive non-use when the husband was a migrant could be very different than when the husband was a resident and staying in a high out-migration environment.” I am not clear on how the study data supports this statement, since findings are not presented separately for migrant males vs. resident males, and the study did not interview women whose husbands were residents. Also, several reasons cited for non-use (cultural norms, misinformation, etc.) are common across other LMIC settings without high out-migration. I think it would be fair to say that the study demonstrates some reasons for contraceptive non-use that are unique to the high out-migration environment, and others that are common across multiple settings.

Reply: Thank you for pointing this out. We have re-written the sentence to make sure it reflects our study findings. 

17. Figure 1: I appreciate that the authors are attempting to visually show the complex relationships among factors contributing to non-use of contraception. However, the indirect pathways shown in this figure are not fully supported by the data. For example, there is a dotted line (indirect link) from “poor couple communication” to “pregnancy as a contraceptive”, “need to prove fertility”, and “male child preference”, but the results section does not provide data indicating these factors are related in some way. If the authors decide to include this figure, I would suggest to clarify that the indirect pathways reflect the authors’ understanding of causal relationships/pathways in the existing literature, rather than empirical data from this study. Alternatively, I think this figure could be removed and the authors could simply state that the factors influencing contraceptive use in high-migration settings a) may differ in some ways from other settings, and b) are complex/nuanced (as is the case in most settings).

Reply: Thank you again for your guidance on this. We have re-articulated to make it clear that the indirect relationships are based on empirical evidence and synthesis of findings. 

18. Line 472: add the word “contraceptive” before “non-use”.

Reply: Suggestion incorporated in the revised manuscript. 

19. Line 477-78: This final sentence feels a bit confusing. How would one put migration at the center of FP programs? Perhaps this could be reworded to say “the unique context of high out-migration settings needs to be considered and addressed in FP programs”

Reply: Thanks for the suggestion. We have incorporated this in the revised manuscript.

20. See above comment about use of the terms “micro” and “macro” – this feels confusing, as these terms usually distinguish individual vs. community/structural factors, and that’s not quite the case here.

Reply: Yes, we have removed them as suggested. 

Response to the reviewer #4: 

1.Methodology is sound, well-explained, thorough, with very good sample size. A couple of items where clarifications are needed: (1) The introduction comments on HIV as a problem among those not using any contraception, but results and conclusion do not discuss barrier methods vs other methods that would not protect against sexually-transmitted disease. (2) There is also not enough comment in the results or discussion on permanent or longer-term (tubal ligation, IUD) vs temporary contraception. Given inability to procure on time before husband's arrival is an issue, one would assume most of the discussions revolved around temporary contraception (pill, condom, etc), but there are mentions of tubal ligations and some fears of IUDs causing health issues. This needs further discussion or clarification.

Reply: Thank you for highlighting some important points. We like to clarify on the two points: (1) In the introduction, we did not discuss HIV among those not using contraceptive. Rather, we provided existing evidence on how migration has been an important factor in Bihar in explaining health situation; (2) Given our interest was to explore the reasons for contraceptive non-use, we feel discussion on use of spacing and limiting method seems to beyond the scope of the paper. On a related note, tubal ligation is a permeant method and women prefer to go for it once they had completed their family size. Moreover, women would prefer to go for sterilization when their husbands would visit home. 

We again thank the reviewers and you for giving us this opportunity and will be happy to provide further clarification if required.

With warm regards

Bidhu

bbmahaptra@gmail.com

---

## [Decision Letter · Decision Letter 1]

8 Feb 2021

PONE-D-20-13937R1

Why women do not use contraceptives: Exploring the role of male out-migration

PLOS ONE

Dear Dr. Mahapatra,

Thank you for submitting your manuscript to PLOS ONE. After careful consideration, we feel that it has merit but does not fully meet PLOS ONE’s publication criteria as it currently stands. Therefore, we invite you to submit a revised version of the manuscript that addresses the points raised during the review process.

Please review the concerns raised by Reviewer 3  The majority of the comments are minor and can be easily addressed.  In addition to addressing the substantive comments, please ensure that the manuscript is carefully reviewed for language, grammar, and punctuation.  Plos One does not provide these services and thus authors are responsible for final editing. 

We look forward to receiving your revised manuscript.

Kind regards,

Linnea A Zimmerman, Ph.D, MPH

Academic Editor

PLOS ONE

Reviewers' comments:

Reviewer's Responses to Questions

**Comments to the Author**

1. If the authors have adequately addressed your comments raised in a previous round of review and you feel that this manuscript is now acceptable for publication, you may indicate that here to bypass the “Comments to the Author” section, enter your conflict of interest statement in the “Confidential to Editor” section, and submit your "Accept" recommendation.

Reviewer #1: All comments have been addressed

Reviewer #3: (No Response)

2. Is the manuscript technically sound, and do the data support the conclusions?

Reviewer #1: Yes

Reviewer #3: Yes

3. Has the statistical analysis been performed appropriately and rigorously? 

Reviewer #1: N/A

Reviewer #3: Yes

4. Have the authors made all data underlying the findings in their manuscript fully available?

Reviewer #1: No

Reviewer #3: No

5. Is the manuscript presented in an intelligible fashion and written in standard English?

Reviewer #1: Yes

Reviewer #3: Yes

6. Review Comments to the Author

Reviewer #1: Thank you for addressing my comments, this is much improved and will be a valuable contribution to the literature.

Reviewer #3: Thank you to the authors for responding to the reviewer comments and revising the manuscript. I have a few additional comments on the revised manuscript:

Overall: since PLOS One does not copyedit accepted manuscripts, the authors need to copyedit the manuscript carefully. This is particularly applicable to text that was added in this revision -- there are some missing articles ("a", "the", etc) and typos.

Study design:

1) The justification for using FGDs still seems unclear, as divergent views can also emerge by interviewing people individually (in fact, IDIs may be more likely to elicit divergent views on sensitive topics because people are more likely to speak freely). Do the authors mean to say that they wanted to capture broader community perceptions and norms on contraceptive use rather than people's individual contraceptive practices?

2) Can the authors please clarify the following statement: "The other objective of conducting FGD was to inform the quantitative survey in developing study tool tuned for study context." What quantitative survey? How would FGD findings be preferable to in-depth interview findings in informing development of a quantitative tool?

3) In their response to the reviewers, the authors explained why they selected two high male out-migration and two low male out-migration blocks in each district, and indicated that this stratification was dropped in the analysis phase as no major differences in themes emerged. However, this explanation was not added to the manuscript text. Please add this explanation.

4) The following sentence is unclear, please revise: "At the end of the study, while from each study group, one FGD was conducted in each block, for men in one block, an additional FGD was conducted to capture the diverging views of population." Are the authors saying that the FGDs didn't elicit many divergent views, so they conducted one additional FGD at the end of the study to try to elicit divergent views?

Results:

1) I am a bit confused about why "poor couple communication on FP" is categorized as a characteristic of the migration environment. This seems directly related to the husband's outmigration, since it means that couples talk infrequently and need to reserve those conversations for essential topics. Perhaps this could be recategorized?

2) "Fear of sickness due to side effects" -- suggest to rename this simply "Fear of side effects", and move the content on side effects up to this section (see next comment).

3) The first paragraph under "myths and misperceptions" describes legitimate side effects that can be experienced with contraceptive method use, as well as misperceptions (i.e. contraceptive methods cause cancer). I would suggest to move the content about fear of side effects to the section above "Fear of sickness due to side effects", and reserve this section for true myths/misperceptions.

Discussion:

1) Page 15: "Whereas, at the system level, low ASHA outreach on FP, need to prove fertility, preference of male child, myths and misconception about contraception, poor couple communication on FP were listed as deterrents to contraceptive use." -- The term "system level" seems appropriate when talking about ASHA outreach and lack of accurate communication on contraception, but the other factors don't quite fit under this term. Suggest to use a more comprehensive term like "system-level and community/cultural factors".

2) Limitations: I recommend that the authors acknowledge the potential drawbacks of using FGDs to discuss sensitive topics like contraception -- namely that people may not feel comfortable to openly express their views in a large group, particularly if their opinion differs from others / the loudest voices in the group.

7. PLOS authors have the option to publish the peer review history of their article (what does this mean?). If published, this will include your full peer review and any attached files.

Reviewer #1: **Yes: **Nadia Diamond-Smith

Reviewer #3: No

---

## [Author Response · Author response to Decision Letter 1]

17 Feb 2021

Response to the reviewer # 3: 

Overall: 

Since PLOS One does not copyedit accepted manuscripts, the authors need to copyedit the manuscript carefully. This is particularly applicable to text that was added in this revision -- there are some missing articles ("a", "the", etc) and typos.

Response: Thank you. We have done a thorough editing of the manuscript. 

Study design:

1. The justification for using FGDs still seems unclear, as divergent views can also emerge by interviewing people individually (in fact, IDIs may be more likely to elicit divergent views on sensitive topics because people are more likely to speak freely). Do the authors mean to say that they wanted to capture broader community perceptions and norms on contraceptive use rather than people's individual contraceptive practices?

Response: Yes, we wanted to capture the broader community perceptions.

2. Can the authors please clarify the following statement: "The other objective of conducting FGD was to inform the quantitative survey in developing study tool tuned for study context." What quantitative survey? How would FGD findings be preferable to in-depth interview findings in informing development of a quantitative tool?

Response: Thank you for highlighting this. FGDs were not only informed in framing the right questions for migration context but also guided the analysis of quantitative survey data. In terms of survey tool development, it helped in framing the questions in local language and picking the right terms to be used in question framing and response categories. In case of In-depth interviews, terms prevalent at community level may not have emerged so clearly.

The quantitative survey was followed by the qualitative phase where the objective was to examine if the contraceptive use behaviours of women in migration affected areas were linked to migration of women’s husband. A paper based on quant data has been published and can be accessed here: https://bmcpublichealth.biomedcentral.com/articles/10.1186/s12889-020-09906-9

3. In their response to the reviewers, the authors explained why they selected two high male out-migration and two low male out-migration blocks in each district, and indicated that this stratification was dropped in the analysis phase as no major differences in themes emerged. However, this explanation was not added to the manuscript text. Please add this explanation.

Response: Thank you for pointing this out. We have included a sentence to this effect in the revised manuscript.

4. The following sentence is unclear, please revise: "At the end of the study, while from each study group, one FGD was conducted in each block, for men in one block, an additional FGD was conducted to capture the diverging views of population." Are the authors saying that the FGDs didn't elicit many divergent views, so they conducted one additional FGD at the end of the study to try to elicit divergent views?

Response: Thank you for point this out. In one of the blocks FGDs for migrant and non-migrant men were collected separately. We meant that the view of these groups was very different. In the revised manuscript, we have clarified this. 

Results:

1. I am a bit confused about why "poor couple communication on FP" is categorized as a characteristic of the migration environment. This seems directly related to the husband's outmigration, since it means that couples talk infrequently and need to reserve those conversations for essential topics. Perhaps this could be recategorized?

Response: Poor couple of communication on FP was prevalent across all couples residing in the migration affected areas irrespective of whether women had migrant and resident husband. Couples generally refrained from discussing about contraceptives amongst themselves. This was emerged from discussion with men and ASHAs as well. We have made some revision to the text to bring more clarity around this.

2. "Fear of sickness due to side effects" -- suggest to rename this simply "Fear of side effects", and move the content on side effects up to this section (see next comment).

Response: We have changed the “Fear of sickness due to side effects” to “Fear of side effects” in the revised manuscript. 

3. The first paragraph under "myths and misperceptions" describes legitimate side effects that can be experienced with contraceptive method use, as well as misperceptions (i.e. contraceptive methods cause cancer). I would suggest to move the content about fear of side effects to the section above "Fear of sickness due to side effects", and reserve this section for true myths/misperceptions.

Response: Thank you again for bringing this to our attention. As suggested, in the revised manuscript we have moved the content about fear of side effects to the section "Fear of side effects” and revised the text on myths and misperceptions section to make it more relevant.

Discussion:

1. Page 15: "Whereas, at the system level, low ASHA outreach on FP, need to prove fertility, preference of male child, myths and misconception about contraception, poor couple communication on FP were listed as deterrents to contraceptive use." -- The term "system level" seems appropriate when talking about ASHA outreach and lack of accurate communication on contraception, but the other factors don't quite fit under this term. Suggest to use a more comprehensive term like "system-level and community/cultural factors".

Response: Thanks for highlighting this. We have made the suggested change in the revised manuscript. 

2. Limitations: I recommend that the authors acknowledge the potential drawbacks of using FGDs to discuss sensitive topics like contraception -- namely that people may not feel comfortable to openly express their views in a large group, particularly if their opinion differs from others / the loudest voices in the group.

Response: We have included the limitation with FGDs in the revised manuscript.

---

## [Editor Report · Decision Letter 2]

15 Mar 2021

Why women do not use contraceptives: Exploring the role of male out-migration

PONE-D-20-13937R2

Dear Dr. Mahapatra,

We’re pleased to inform you that your manuscript has been judged scientifically suitable for publication and will be formally accepted for publication once it meets all outstanding technical requirements.

Kind regards,

Linnea A Zimmerman, Ph.D, MPH

Academic Editor

PLOS ONE
---

## [Editor Report · Acceptance letter]

17 Mar 2021

PONE-D-20-13937R2 

Why women do not use contraceptives: Exploring the role of male out-migration 

Dear Dr. Mahapatra:

I'm pleased to inform you that your manuscript has been deemed suitable for publication in PLOS ONE. Congratulations! Your manuscript is now with our production department. 

Kind regards, 

on behalf of

Dr. Linnea A Zimmerman 

Academic Editor

PLOS ONE